# Plastid Phylogenomic Insights into the Inter-Tribal Relationships of Plantaginaceae

**DOI:** 10.3390/biology12020263

**Published:** 2023-02-07

**Authors:** Pingxuan Xie, Lilei Tang, Yanzhen Luo, Changkun Liu, Hanjing Yan

**Affiliations:** 1College of Traditional Chinese Medicine, Guangdong Pharmaceutical University, Guangzhou 510006, China; 2Key Laboratory of Bio-Resources and Eco-Environment of Ministry of Education, College of Life Sciences, Sichuan University, Chengdu 610065, China; 3Key Laboratory of State Administration of Traditional Chinese Medicine for Production and Development of Cantonese Medicinal Materials, Guangzhou 510006, China

**Keywords:** plastome, phylogeny, Plantagineae, Digitalideae, Veroniceae, Hemiphragmeae, Sibthorpieae

## Abstract

**Simple Summary:**

The current classification of Plantaginaceae divides the family into 12 tribes. However, the inter-tribal relationships of this family remain unresolved, and phylogenetic and comparative studies on Plantaginaceae plastomes are also scarce. In this study, we compared 41 plastomes representing the 11 tribes of Plantaginaceae and uncovered the high conservation of the plastomes in terms of genome structure and gene content. Phylogenetic analyses based on 68 plastid protein-coding genes (PCGs) successfully elucidated the inter-tribal relationships of Plantaginaceae, especially disputed tribes, such as Plantagineae, Digitalideae, Veroniceae, Hemiphragmeae and Sibthorpieae. PhyParts analysis showed a certain extent of conflict between gene trees and species tree, revealing the limitations of phylogenetic analysis of Plantaginaceae using single or multiple plastid DNA sequences. Collectively, our results provide some basic information on the plastomes of the Plantaginaceae taxa and some new insights into the inter-tribal relationships of Plantaginaceae, laying the groundwork for future phylogenetic and taxonomic studies.

**Abstract:**

Plantaginaceae, consisting of 12 tribes, is a diverse, cosmopolitan family. To date, the inter-tribal relationships of this family have been unresolved, and the plastome structure and composition within Plantaginaceae have seldom been comprehensively investigated. In this study, we compared the plastomes from 41 Plantaginaceae species (including 6 newly sequenced samples and 35 publicly representative species) representing 11 tribes. To clarify the inter-tribal relationships of Plantaginaceae, we inferred phylogenic relationships based on the concatenated and coalescent analyses of 68 plastid protein-coding genes. PhyParts analysis was performed to assess the level of concordance and conflict among gene trees across the species tree. The results indicate that most plastomes of Plantaginaceae are largely conserved in terms of genome structure and gene content. In contrast to most previous studies, a robust phylogeny was recovered using plastome data, providing new insights for better understanding the inter-tribal relationships of Plantaginaceae. Both concatenated and coalescent phylogenies favored the sister relationship between Plantagineae and Digitalideae, as well as between Veroniceae and Hemiphragmeae. Sibthorpieae diverged into a separate branch which was sister to a clade comprising the four tribes mentioned above. Furthermore, the sister relationship between Russelieae and Cheloneae is strongly supported. The results of PhyParts showed gene tree congruence and conflict to varying degrees, but most plastid genes were uninformative for phylogenetic nodes, revealing the defects of previous studies using single or multiple plastid DNA sequences to infer the phylogeny of Plantaginaceae.

## 1. Introduction

Plantaginaceae, a family of approximately 90 genera and 1900 species with great morphological diversity, includes annual or perennial herbs, sometimes shrubs (e.g., *Veronica* sect. *Hebe*, *Aragoa*), aquatics (e.g., *Callitriche*, *Hippuris*, etc.) or even carnivorous plants (e.g., *Philcoxia*) [1]. Many taxa (e.g., *Plantago*, *Digitalis*, *Antirrhinum*, etc.) have been widely explored and found to be of great medicinal, ornamental, and economic value. Since the clade “scroph Ⅱ” was discovered by Olmstead and Reeves [2], it has received considerable attention from researchers. Ultimately, a newly expanded clade now formally called Plantaginaceae, also accepted by APG IV [3], emerged from the former Scrophulariaceae due to the results of molecular phylogenetic studies [4,5,6,7,8,9,10]. According to the classification of Albach et al. [7], the enlarged Plantaginaceae contains 12 tribes (Plantagineae, Veroniceae, Digitalideae, Globularieae, Hemiphragmeae, Sibthorpieae, Russelieae, Cheloneae, Gratioleae, Antirrhineae, Callitricheae and Angelonieae) and each tribe was resolved as a well-supported monophyletic lineage [7,11,12,13,14].

Previous phylogenetic studies on Plantaginaceae were mainly based on single or multiple-locus DNA sequence data, such as nuclear ribosomal DNA internal-transcribed spacer (ITS), external-transcribed spacer (ETS), and plastid DNA *ndh*F, *rbc*L, *trn*L-*trn*F, *rps*16 intron and *mat*K-*trn*K intron regions [7,11,12,15], providing some insights into phylogenetic relationships within Plantaginaceae. Nevertheless, these analyses failed to sort out the relationships among tribes well. For instance, the phylogenetic relationships among Plantagineae, Veroniceae, Digitalideae, Hemiphragmeae, Sibthorpieae, Cheloneae and Russelieae were unclear on account of weak support or even low resolution, which was shown in detail by Albach et al. [7] and systematically reviewed by Tank et al. [16]. Similar problems also arose in follow-up studies [10,11,12,13]. Additionally, incongruent topologies often appeared in Plantaginaceae phylogenetic trees, resulting in the phylogenetic positions of these tribes, such as Hemiphragmeae, Digitalideae, Plantagineae and Veroniceae, remaining ambiguous [7,11,12]. Specifically, there was an indication for a sister-group relationship of Hemiphragmeae and the clades of Digitalideae, Plantagineae and Veroniceae with strong support [7], whereas Hemiphragmeae was strongly supported as a sister to Veroniceae by Gormley et al. [12].

Recently, Mower et al. [17] suggested that Plantagineae may be more closely related to Digitalideae, while Veroniceae are related to Hemiphragmeae. The sister-group relationship of Plantagineae and Digitalideae was reproduced in their subsequent studies [14]. However, these studies mainly shed light on the relationships among genera in Plantagineae rather than tribes within Plantaginaceae. In addition, the taxon sampling of the tribes involved was also insufficient, thus still requiring in-depth analysis to uncover the phylogenetic relationships among these tribes. Overall, the limited taxon sampling and datasets consisting of single or a few DNA sequences may lead to an erroneous phylogenetic tree [18,19]. Therefore, to clarify the tribal-level phylogenetic relationships of Plantaginaceae, it is necessary to reconstruct a robust phylogeny based on a broad sampling and integration of more genes or genomic scale sequence data.

With the development of next-generation sequencing (NGS) technologies, plastid genomes (plastomes) have become more accessible. Owing to usually uniparental inheritance (in most case maternal inheritance), lack of recombination and moderate evolutionary rate [20,21], plastomes have been widely used for phylogenetic inferences [22,23,24]. In addition, structural variation, gene content and gene arrangement among plastomes can also serve as a supplement for phylogenetic inferences [25,26].

In this study, we newly sequenced the complete plastomes of six Plantaginaceae species and compared them with 35 publicly available plastomes of Plantaginaceae from GenBank. To investigate inter-tribal relationships in Plantaginaceae, a dataset of 43 species (41 Plantaginaceae species and two outgroup species) covering 11 tribes (except Globularieae) was constructed using 68 plastid protein-coding genes. Apart from concatenated analyses, we also undertook phylogenetic analysis using ASTRAL-III [27] as well as methods to assess the extent of conflict/concordance among plastid gene trees across the species tree [28]. The objectives of our study are to (1) investigate the structural and compositional variation of plastomes within Plantaginaceae; (2) reconstruct a robust phylogeny and preliminarily elucidate the relationships of the disputed tribes Plantagineae, Veroniceae, Digitalideae, Hemiphragmeae and Sibthorpieae; (3) assess the degree of gene-tree concordance and conflict within plastid data of Plantaginaceae, revealing the limitation of phylogenetic analyses using single or multiple plastid DNA sequences.

## 2. Materials and Methods

### 2.1. Plant Materials, DNA Extraction, Sequencing, Assembly and Annotation

A dataset of plastomes from 43 species was used in this study, representing 11 tribes of Plantaginaceae (except Globularieae). Among them, six species (*Trapella sinensis*, *Adenosma glutinosum*, *Deinostema violacea*, *Ellisiophyllum pinnatum*, *Russelia equisetiformis* and *Linaria buriatica*) were newly sequenced in this study. All voucher specimens were deposited in the Herbarium of Kunming Institute of Botany (KUN), Chinese Academy of Sciences. Voucher information, the source of material and GenBank accession numbers are given in Appendix A. Pastomes of the other 37 species (including 2 outgroup taxa) were obtained from GenBank of the National Center for Biotechnology Information (NCBI) Appendix A).

Genomic DNA of each sample was extracted from the silica gel-dried leaf tissues using the modified CTAB method [29]. Paired-end libraries with an average insert size of approximately 400 bp were prepared using a TruSeq DNA Sample Prep Kit (Illumina, Inc., San Diego, CA, USA) according to the manufacturer’s instructions. The libraries were sequenced on the Illumina NovaSeq platform at Personalbio (Shanghai, China). Raw data were filtered using fastP v0.15.0 (-n 10 and -q 15) to obtain high-quality reads [30]. Clean paired-end (PE) reads were assembled using GetOrganelle v1.7.1 [31] with K-mer values of 21, 45, 65, 85, and 105 as seen using SPAdes v3.10 [32]. The complete plastome of *Bacopa monnieri* (MN736955) was set as the reference for *Trapella sinensis*, *Adenosma glutinosum*, and *Deinostema violacea*, while *Antirrhinum majus* (MW877560) was used for assembly the plastomes of *Linaria buriatica*, *Russelia equisetiformis*, and *Ellisiophyllum pinnatum.* Bandage v0.8.1 [33] was used to visualize and filter out the assembled contigs to generate a complete circular plastome. Plastid Genome Annotator [34] was used to annotate the newly assembled plastomes and then manually check the consistency of start/stop codons against the reference genome in Geneious v10.2.3 [35]. The tRNAscanSE web service [36] was used to confirm the tRNA genes with default parameters. Furthermore, the online program OrganellarGenomeDRAW (OGDRAW) [37] was used to draw circular plastome maps. GenBank accession numbers of the six plastomes generated in this study are presented in Appendix A.

### 2.2. Plastome Comparative Analyses

To explore the structural variation of plastomes in Plantaginaceae, a set of 15 representative species was ultimately chosen, representing Plantagineae (*Plantago fengdouensis*, *P. media*, *P. maritima*, *P. nubicola* and *Littorella uniflora*), Veroniceae (*Veronica persica*), Digitalideae (*Digitalis lanata*), Hemiphragmeae (*Hemiphragma heterophyllum*), Sibthorpieae (*Ellisiophyllum pinnatum*), Callitricheae (*Callitriche palustris*), Russelieae (*Russelia equisetiformis*), Antirrhineae (*Antirrhinum majus*), Cheloneae (*Penstemon rostriflorus*), Gratioleae (*Scoparia dulcis*) and Angelonieae (*Angelonia angustifolia*). IR expansion/contraction of these plastomes was checked using IRscope [38]. After removing one copy of the IR region for each plastome, the general structural features were also analyzed with Mauve v2.3.1 [39].

### 2.3. Phylogenetic Analyses

Typically, organellar genomic regions are concatenated as a single locus to reconstruct phylogeny. However, a recent study uncovered notable levels of gene-tree conflict within the plastome [40], suggesting that concatenated analyses may be inappropriate for plastomes and should be performed with caution. In light of this, Gonçalves et al. [41] advocated for the use of both concatenation and multi-species coalescent methods (MSC) for inferring plastid phylogeny. Multispecies coalescent algorithms are divided into summary methods that use estimated gene trees to infer the species tree and “single-site” methods that use nucleotide alignments for species tree inference. Despite not performing full coalescent analysis, the summary method ASTRAL [27], used to estimate a species tree given a set of gene trees, has been shown to be statistically consistent under the multi-species coalescent model [42]. Therefore, we utilized both concatenated and ASTRAL analyses to infer phylogenetic relationships.

Before phylogenetic analyses, all 68 protein-coding genes (PCGs) were extracted from the plastomes of 43 taxa (Appendix A) using Geneious v10.2.3 [35]. Each protein-coding gene was separately aligned using MAFFT v7.450 [43] with the default settings and adjusted manually. Characteristics of alignments of 68 PCGs involved in the phylogenetic analyses are listed in Appendix A. *Scrophularia dentata* and *Buddleja sessilifolia* (Scrophulariaceae: Lamiales) served as the outgroup according to the results of Gormley et al. [13].

For concatenated analysis, the multiple alignments of genes were concatenated into a 61,841 bp matrix (Appendix A) with a preparation of a partition file generated by PhyloSuite v1.2.2 [44]. Maximum likelihood (ML) analyses were performed by IQ-TREE v2.1.2 [45] using UFBoot2 [46] with the most suitable partition models (Appendix A) found by Modelfinder [47] with 1000 replicates. Besides, the SH-aLRT test [48] was used to assess branch supports. According to the Akaike information criterion (AIC), PartitionFinder v2.1.1 [49] was used to select the best-fit partitioning schemes and models for each gene with the default values (Appendix A). Bayesian inference (BI), implemented in MrBayes v3.2.7a [50], was constructed with an average deviation of split frequencies below 0.01. Approximately 1000,000 generations were conducted for the matrix, and each set was sampled every 2500 generations with a burn-in of 25%. The software FigTree 1.4.4 (http://tree.bio.ed.ac.uk/software/Figtree/ (accessed on 1 September 2022)) was used to visualize the phylogenetic tree.

For the ASTRAL analysis, 68 plastid gene trees were inferred separately using RAxML v8.2.11 [51] with the GTRGAMMA model and 100 bootstrap replicates, and low-supported branches (<10% bootstrap support) in the gene trees were collapsed to improve species-tree inference [52]. Subsequently, the collapsed gene trees were imported into ASTRAL-III [27] with the default settings, which produced an estimated topology of the species tree with branch lengths and local posterior probabilities (LPP) as branch support values.

In addition, PhyParts [28] was used to examine how individual gene trees agree/conflict with the species tree by mapping the 68 plastid gene trees onto the species tree generated by ASTRAL analysis, with bootstrap support (BS) < 70% at gene-tree branches considered uninformative. A Python script by M. Johnson (https://github.com/mossmatters/phyloscripts/tree/master/phypartspiecharts (accessed on 20 October 2022)) was used to visualize the output of the PhyParts analyses. Pie charts on the species phylogeny show the number of gene trees that were concordant, conflicting, or uninformative concerning each node in the species tree.

## 3. Results

### 3.1. Features of the Plastomes

Illumina sequencing generated 6,069,200–30,408,958 paired-end clean reads for the six samples. Among them, 302,408–5,093,971 reads were mapped to the final assembly, with the average coverage ranging from 295.979× to 4,997.812× (Appendix A). All newly sequenced plastomes were assembled into a typical quadripartite structure containing a large single copy (LSC) and a small single copy (SSC) separated by two inverted repeats (IRs, including IRa and IRb) (Figure 1).

By comparison, the size of the plastomes in Plantaginaceae ranged from 130,833 bp (*Littorella uniflora*) to 165,045 bp (*Plantago asiatica*) (Table 1), which identically consist of a pair of IRs (IRa and IRb, with lengths of 21,404–38,724 bp) separated by a LSC region (77,490–85,713 bp) and a SSC region (4577–18,447 bp) (Table 1). The total GC content ranged from 37.4% to 39.0%, with the lowest GC in *Trapella sinensis* and *Limnophila sessiliflora* and the highest GC content in *Littorella uniflora* (Table 1). Most plastomes contained the same 114 unique genes, including 80 protein-coding genes, 30 tRNA genes and 4 rRNA genes (counting all duplicated genes only once) (Table 1, Appendix A). Notably, the *ycf*15 gene is lost from the plastomes of *Plantago*, *Littorella uniflora*, *Veronica polita*, *Veronica persica*, *Veronica eriogyne* and *Angelonia angustifolia*. In addition, *Littorella uniflora* lacks functional copies of *ndh* genes (except *ndh*E); *Angelonia angustifolia* and *Scoparia dulcis* lack the *inf*A gene, and a novel tRNA gene (*trn*L-CAG) is present in plastomes of *Plantago maritima* (Appendix A).

### 3.2. Plastome Structural Variation

The IRs/LSC and IRs/SSC junctions among the plastomes of 15 diverse species representing 11 tribes in Plantaginaceae were compared to identify IR expansion/contraction. Overall, the boundaries of the LSC/IRb, IRb/SSC, SSC/IRa and IRa/LSC are relatively conservative within Plantaginaceae plastomes, with the *rps*19, *ndh*F, *ycf*1 and *trn*H genes located in the above junctions, respectively (Figure 2). The collinearity analysis also showed the conservation of plastome gene arrangement in most tribes (Figure 3). In contrast, three *Plantago* species, *Plantago fengdouensis*, *P. media* and *P. maritima,* have distinct IR expansions, as well as rearrangements, in the SSC and IRs, which leads to differences in the junction and gene arrangement (Figure 2 and Figure 3 and Appendix A). In addition, the size reduction in the LSC, SSC, and IRs of *Littorella uniflora*, especially the loss of *ndh* genes, brings about changes in the IRb/SSC junction (Figure 2).

### 3.3. Phylogenetic Analyses

Based on the concatenated dataset, both ML and BI phylogenies generated identical tree topologies, with nearly all nodes exhibiting high support values (ultrafast bootstrap (UFboot) = 100; SH-aLRT values (SH-aLRT) = 100; posterior probability (PP) = 1.00) (Figure 4), which recovered six distinct clades within Plantaginaceae: (Ⅰ) Gratioleae + Angelonieae (UFboot = 100; SH-aLRT = 100; PP = 1), (Ⅱ) Russelieae + Cheloneae (UFboot = 100; SH-aLRT = 100; PP = 1.00), (Ⅲ) Antirrhineae (UFboot = 100; SH-aLRT = 100; PP = 1.00), (Ⅳ) Callitricheae (UFboot = 100; SH-aLRT = 100; PP = 1.00), (Ⅴ) Sibthorpieae (UFboot = 100; SH-aLRT = 99.7; PP = 1.00), and (Ⅵ) Plantagineae + Digitalideae, Veroniceae + Hemiphragmeae (UFboot = 100; SH-aLRT = 100; PP = 1.00) (Figure 4). Among them, Gratioleae was clustered with Angelonieae at the base of the phylogenetic tree (Clade Ⅰ) sister to the rest of Plantaginaceae. Within the remaining tribes, Russelieae and Cheloneae formed the early diverging Clade Ⅱ, which was successively sister to Clade Ⅲ (Antirrhineae), Clade Ⅳ (Callitricheae), Clade Ⅴ (Sibthorpieae), and the Clade Ⅵ, in which Plantagineae was sister to Digitalideae (UFboot = 94; SH-aLRT = 95.3; PP = 1.00), while Veroniceae was supported as sister to Hemiphragmeae (UFboot = 100; SH-aLRT = 100; PP = 1.00).

Despite slight differences in some relationships observed within Veronica and Penstemon, the species tree topology implemented in ASTRAL was basically congruent with the result of the concatenation analyses, with high local posterior probability (LPP) values at most nodes (Figure 5). Notably, support values at the nodes of Digitalideae + Plantagineae (LPP = 0.7) and Russelieae + Cheloneae (LPP = 0.78) were relatively low, while the location of Sibthorpieae (LPP = 0.87) was only moderately supported.

### 3.4. Concordance and Conflict of the Gene Tree

PhyParts analyses revealed the numbers of consistent, conflicting, or uninformative gene trees for each node in the species tree, with a bootstrap support (BS) threshold of 70 set at nodes of each gene tree (Figure 6). From the pie chart, although a certain number of genes exhibited concordance, the majority of the plastid genes were largely uninformative for nodes of the phylogeny. Besides, there are gene-tree conflicts for nodes of the species tree of varying degrees. Notably, some nodes with the conflicting genes numbers close to or greater than the numbers of concordant genes seemed to have relatively low support in the species tree. The node of Digitalideae and Plantagineae exhibited more gene-tree conflict, with two concordant genes and five conflicting genes, while the node of Sibthorpieae had five concordant genes and ten conflicting genes. Besides, only five genes supported the sister relationship between Cheloneae and Russelieae, while thirteen genes conflicted with the topology (Figure 6).

## 4. Discussion

In general, the plastomes of Plantaginaceae are largely conserved in structure and gene content. Significant IR expansion and rearrangement of plastomes in *Plantago* were observed in three representative species including *Plantago fengdouensis*, *P. media* and *P. maritima*. Indeed, IR expansion and rearrangement of plastomes in *Plantago* have been discussed in detail by Mower et al. [17], and not discussed at length here. Furthermore, the absence of the *ndh* genes, *ycf*15 gene and *inf*A gene found in Plantaginaceae appears to be common in angiosperms [53,54,55,56]. Gene loss can be the consequence of a sudden mutational event or the result of a slow process of accumulation of mutations during the pseudogenization that follows an initial loss-of-function mutation [57]. The loss of *ndh* genes only occurs in *Littorella uniflora*, which may be related to its amphibious lifestyle and partial reliance on Crassulacean acid metabolism (CAM) photosynthesis [17]. The *ycf*15 gene is lost in *Plantago*, *Littorella uniflora*, *Veronica polita*, *V. persica*, *V. eriogyne* and *Angelonia angustifolia*. Indeed, loss of the *ycf*15 gene has been observed in a variety of angiosperm lineages, which may have occurred independently throughout the evolution of angiosperms [54,58,59]. *Angelonia angustifolia* and *Scoparia dulcis* lack the *inf*A gene and in many cases, the loss of the *inf*A gene has been considered to be independently shifted to and expressed in the nucleus [55,60].

Given the weak support and conflicting topology in the phylogenetic tree, previous phylogenetic analyses based on single or several different plastid DNA sequences have failed to clarify the tribe-level relationships within Plantaginaceae [7,11,12,13], especially the phylogenetic placements of Russelieae, Cheloneae, Sibthorpieae, Digitalideae, Plantagineae, Veroniceae, and Hemiphragmeae. Indeed, the results of our concordance and conflict analysis verify these limitations. As shown in Figure 6, due to the lack of sufficient phylogenetic signal [40], most plastid genes are considered uninformative for nodes (BS < 70%) in species tree phylogeny. In addition, against a backdrop of largely uninformative genes, a few genes exhibit well-supported conflict at nodes to varying degrees, which can also correspond to the inter-tribal conflict in previous studies.

In most cases, systematic and stochastic error (presence of a non-phylogenetic signal in the data; the length of the genes) has been invoked by multiple studies to explain the potential source of conflict across the plastome [40,61,62,63]. Additionally, biparental inheritance and heteroplasmic recombination (both intra- and interspecific) have also been proposed as the potential source of biological conflict [40,64,65,66,67,68]. Here, we cannot conclude the causes of the well-supported conflict observed at nodes in our phylogeny based on the present limited analyses, which is also beyond the scope of this study. However, significantly, nodes with apparent conflicts require further research into the causes of conflict, particularly the possibility of biological conflict [40], which might help to understand the evolutionary history of plastomes of related taxa in Plantaginaceae. Moreover, the non-phylogenetic signals which can swamp the faint genuine phylogenetic signal presented in phylogeny also need to be appreciated [61,62].

In contrast, our phylogenomic analyses recovered the phylogenetic relationships of 11 tribes (except Globularieae) with robust support based on 68 plastid PCGs, providing new insights for better understanding the inter-tribal relationships of Plantaginaceae. On the whole, the phylogenetic positions of Gratioleae, Angelonieae, Russelieae, Cheloneae, Antirrhineae and Callitricheae are largely consistent with previous studies [7,8,10,11,12,13,14,15] and are strongly supported in our study. Therefore, the relationship between these tribes is not too controversial. Notably, the sister relationship between Russelieae and Cheloneae was recovered with robust support (UFboot = 100; SH-aLRT = 100; PP = 1.00), which was unprecedented in previous analyses [7,8,10,12,13,15]. Despite the substantial morphological differences between Russelieae and Cheloneae, pair-flowered Cymes (PFCs) were indicated as the common characteristic present in the ancestors of both tribes [69]. Wolfe et al. [15,70] also definitely argued that the cymose inflorescence of Russelieae–Cheloneae would be a synapomorphy of the two tribes.

The placement of Sibthorpieae was elusive in the phylogenetic tree based on different sequence data [7,13]. However, our plastid phylogenomic analyses showed that Sibthorpieae is more closely related to the clade of Plantagineae, Digitalideae, Veroniceae, and Hemiphragmeae with strong support (UFboot = 100; SH-aLRT = 99.7; PP = 1.00), which was not proposed by previous studies. Sibthorpieae comprises the two small genera *Sibthorpia* and *Ellisiophyllum* [7]. Floral morphology with four stamens, a five-lobed corolla, and a five-lobed calyx in Sibthorpieae (mainly in *Ellisiophyllum*) might show a little affinity with some species of its sister clade, particularly *Hemiphragma* and possibly *Wulfenia* of Veroniceae [71]. In addition, the morphological features of pollen grains of Sibthorpieae appear to be transitional. The types of exine sculptures of pollen grains in Sibthorpieae are not only common in most of the species in Russelieae, Cheloneae and Antirrhineae, but are also typical for Plantagineae and Veroniceae [13,72,73,74,75]. Chemically, although Sibthorpieae lacks iridoids and simple phenylethanoids widespread in Plantaginaceae, some of the caffeoyl phenylethanoid glycosides (GPGs) present in Sibthorpieae are similar to those found in Plantagineae, Digitalideae and Veroniceae [76]. Therefore, the placement of Sibthorpieae seems plausible in our study. Nonetheless, the new placement of Sibthorpieae should be treated with caution due to not sampling tribe Globularieae, which used to be related to a clade containing Plantagineae, Digitalideae, Veroniceae and Hemiphragmeae [7,8,11,13,77].

Plantagineae, Veroniceae, Digitaleae and Hemiphragmeae formed the Clade Ⅵ in our phylogenetic analyses. However, the inter-tribal relationships within this clade differed considerably between analyses. Both Olmstead et al. [77] and Bello et al. [8] recovered Digitalideae and Hemiphragmeae as successive sister groups to Veroniceae + Plantagineae with marginal support, whereas Albach et al. [7] found that the phylogenetic location of Digitalideae and Hemiphragmeae interchanged with moderate support. Moreover, Estes and Small [11] identified Digitalideae and Plantagineae as successive sister groups to Veroniceae + Hemiphragmeae, which was also supported by Gomley et al. [12], but the placement of Plantagineae was not well-supported in their studies. Unlike most previous studies, our analyses favor a sister-group relationship between Plantagineae and Digitalideae, and Veroniceae is closely allied to Hemiphragmeae. These affinities are also consistent with recent studies based on plastome, mitochondrial and nuclear ribosomal data [14,17].

Plantagineae was shown to be sister to Veroniceae in most molecular analyses [7,8,77,78], and their shared tetramerous flowers, seed morphology and some similar iridoid constituents between these two tribes were often discussed to show their affinities [7,78]. However, given that the pentamerous and zygomorphic flowers could be plesiomorphic in Plantaginaceae [79], the presence of tetramerous flowers in Plantagineae and Veroniceae was assumed to be an independent shift based on mixed evidence for fusion and loss of flower parts [5,71,78,79,80]. In addition, some similar phytochemical constituents in Plantagineae and Veroniceae seem to be more widespread in Plantaginaceae [77].

Plantagineae encompasses *Plantago*, *Littorella* and *Aragoa* [16], and the (nearly) actinomorphic, tetramerous corolla and the four equal stamens are considered synapomorphies for this tribe [7,78]. In contrast, Digitalideae, including *Digitalis* and *Erinus*, may retain a plesiomorphic character state with a shared long-tubed zygomorphic corolla with five lobes, a deeply five-lobed calyx, and four didynamous stamens (the fifth one is evident only in early ontogeny) [7,78]. Although there are few floral morphological commonalities between Plantagineae and Digitalideae, a similar seed morphology between *Erinus* and Plantagineae and a five-lobed calyx shared in Digitalideae and *Aragoa* might show their affinity [5,78]. Furthermore, similarities in chemosystematic characters could also help to understand their close relationship. Sorbitol, a carbohydrate with a limited distribution, appears to be the sugar characteristic for Digitalideae and Plantagineae, although *Erinus* contains glucose [76,81]. Indeed, the sorbitol found in *Digitalis* is also present in all species of *Plantago* [82] and *Aragoa* [83]. In contrast, mannitol was found to be the sugar characteristic of Veroniceae [76]. In addition, although *Digitalis* contains unique cardenolides and lacks iridoids, *Erinus* preserves the pattern of iridoid biosynthesis, retaining a phytochemical arsenal similar to that of related Plantagineae [76,81].

The sister relationship between Hemiphragmeae and Veroniceae was strongly supported in our analyses, which was also recovered by Gomley et al. [12] and Estes and Small [11], but it was rarely emphasized before. Hemiphragmeae merely contains a monotypic genus *Hemiphragma* from the Himalayas to China, Formosa, Philippines and Celebes with a fleshy, septicidal capsule, four stamens, an actinomorphic five-lobed corolla, a five-lobed calyx, and dimorphic leaves [1,71]. By comparison, Veroniceae with worldwide distribution comprises a large genus *Veronica* and the wulfenioid grade including *Veronicastrum*, *Lagotis*, *Wulfenia*, *Picrorhiza*, *Wulfeniopsis* and *Paederota* [6,7,16]. Almost all genera are characterized by two stamens (except *Picrorhiza*, which has four stamens). *Veronica sensu lato* is mainly characterized by (almost) actinomorphic flowers, a short to absent corolla tube, four-lobed corolla and four-lobed calyx, whereas the majority of genera in the wulfenioid grade have long-tubed zygomorphic flowers, a two-lipped corolla (commonly four corolla lobes) and five-lobed calyx (except *Lagotis,* variable in corolla and calyx number, and *Wulfenia* with a five-lobed corolla) [6]. Obviously, the floral morphological differences between Veroniceae and Hemiphragmeae are therefore unlikely enough to relate these two tribes. Nevertheless, their sister relationship can be justified by the following lines of evidence. First, Hemiphragmeae and Veroniceae share a common pollen type with similar patterns of sculpture of exine and aperture membranes [73]. Next, despite the differences in floral symmetry, the pentamerous calyx and corolla shared in *Wulfenia* and *Hemiphragma* might show a little affinity between the two tribes. Indeed, with several characteristics such as a two-lobed stigma, pentamerous calyx and corolla, divergent anther thecae, and colporate pollen, *Wulfenia* is frequently considered the most primitive representative in Veroniceae [6,84,85]. Furthermore, *Wulfenia*, *Lagotis* and *Veronicastrum* are supposed to be most likely to form the representatives of the early diverging branch of Veroniceae in most molecular studies [4,6,7], and some of the chemical compounds present in *Wulfenia* and *Lagotis* are very similar to those found in *Hemiphragma* [76]. Finally, based on comprehensive analyses [4,86,87], the possible common ancestral distribution area (the Qinghai-Tibetan Plateau (QTP) or Himalayan region) of the two tribes may also provide evidence for their sister relationships. Nonetheless, further studies are necessary to validate the relationships found here.

## 5. Conclusions

The present study enriches the genomic resources of the Plantaginaceae plastome and provides basic information on the plastome structure and gene content of Plantaginaceae taxa. Although Globularieae was not included in our analyses, the reconstructed robust plastome phylogeny, including 11 tribes of Plantaginaceae, provides some new insights into the inter-tribal relationships of Plantaginaceae. In addition, to further clarify the relationships between tribes in Plantaginaceae, future studies will benefit from using more extensive taxonomic sampling of all tribes from the plastid genome and employing more data from the nuclear and mitochondrial genomes.

## Figures and Tables

**Figure 1 biology-12-00263-f001:**
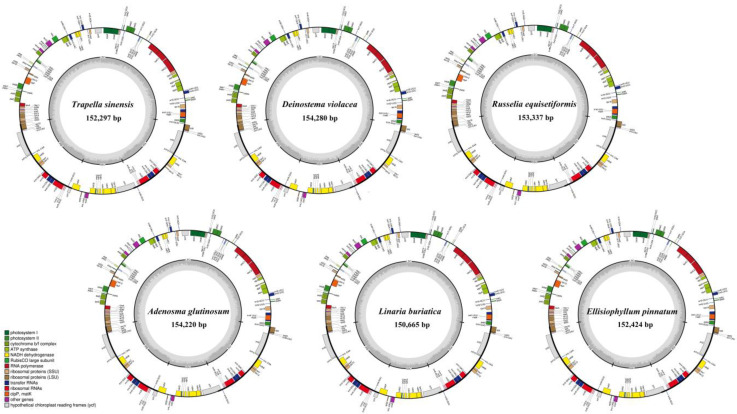
Circular maps of six Plantaginaceae plastomes. Genes shown outside of the outer layer circle are transcribed clockwise, whereas those insides are transcribed counterclockwise. The genes belonging to different functional groups are color-coded. The darker gray area of the inner circle denotes the GC content, while the lighter gray area indicates the AT content of the genome. LSC, large single-copy; SSC, small single-copy; IR, inverted repeat.

**Figure 2 biology-12-00263-f002:**
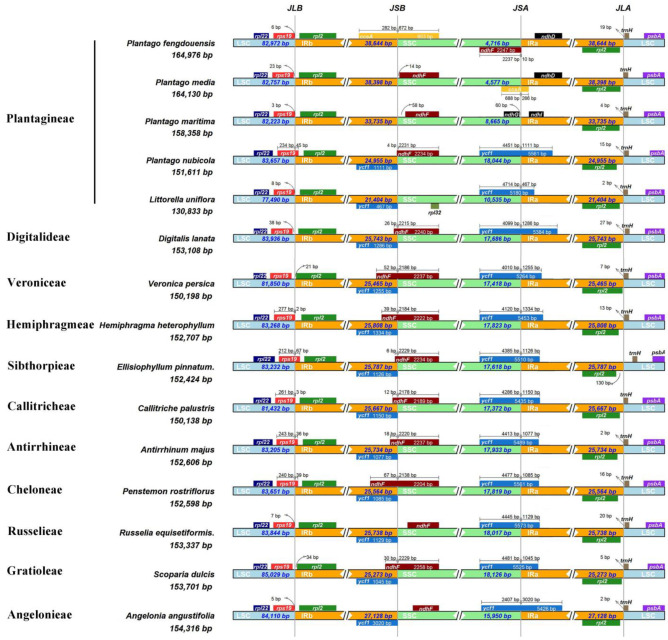
Comparison of the borders of the LSC, SSC, and IR regions of the plastomes from 15 representative species of Plantaginaceae. JLB: junction line between LSC and IRb; JSB: junction line between IRb and SSC; JSA: junction line between SSC and IRa; JLA: junction line between IRa and LSC.

**Figure 3 biology-12-00263-f003:**
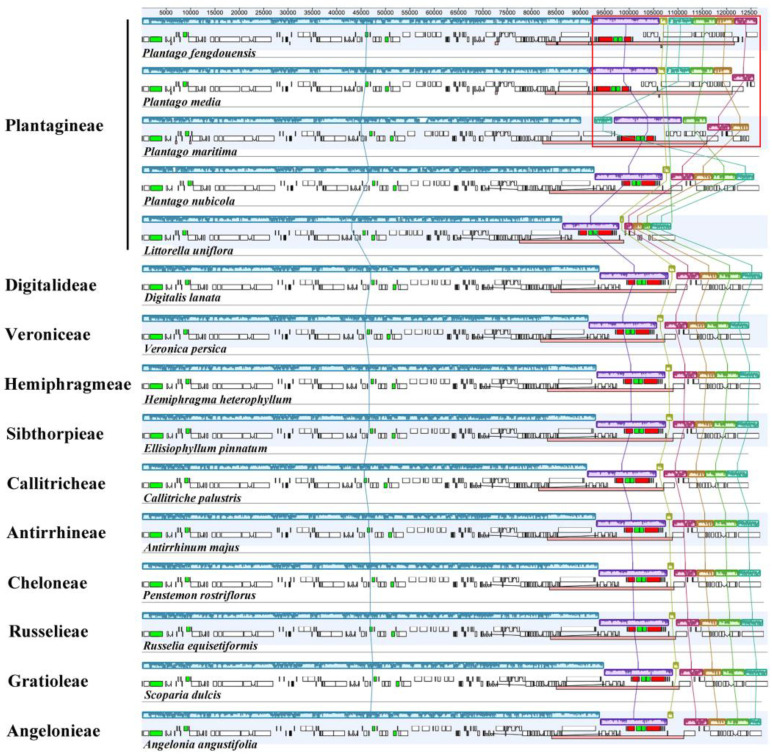
Comparison of plastomes from 15 representative species of Plantaginaceae using the MAUVE algorithm. Rectangular blocks of the same color indicate collinear regions of sequences and the histograms within each block indicate the degree of sequence similarity. The pink blocks indicate the IR regions. The red frame indicates the areas of structural rearrangement, including translocations and inversions (see Appendix A for more details).

**Figure 4 biology-12-00263-f004:**
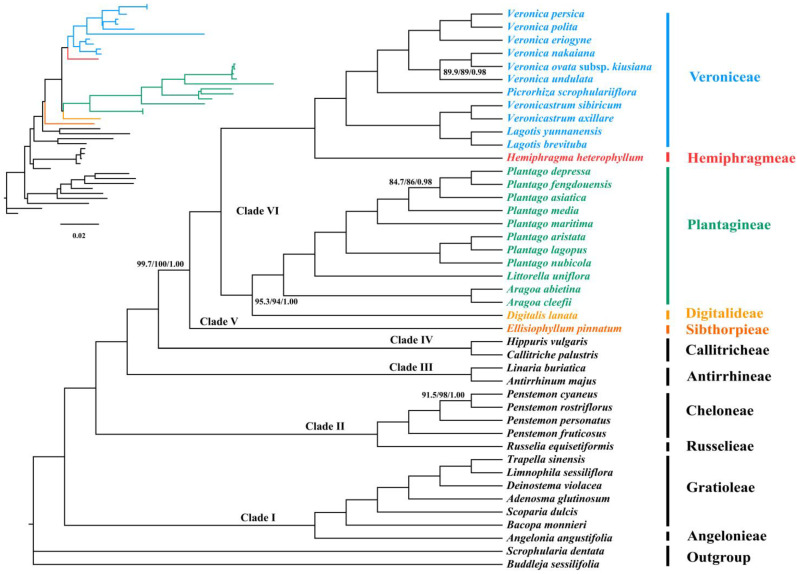
Phylogeny of Plantaginaceae reconstructed by analyses of 68 concatenated plastid PCGs using maximum likelihood (ML) and Bayesian inference (BI) methods. Values at nodes represent the ultrafast bootstrap support values (UFBoot) and SH-aLRT values (SH-aLRT) of the maximum likelihood analysis and the posterior probabilities (PP) of the BI analysis values. All unlabeled nodes indicate high support values (UFBoot = 100; SH-aLRT = 100; PP = 1.00).

**Figure 5 biology-12-00263-f005:**
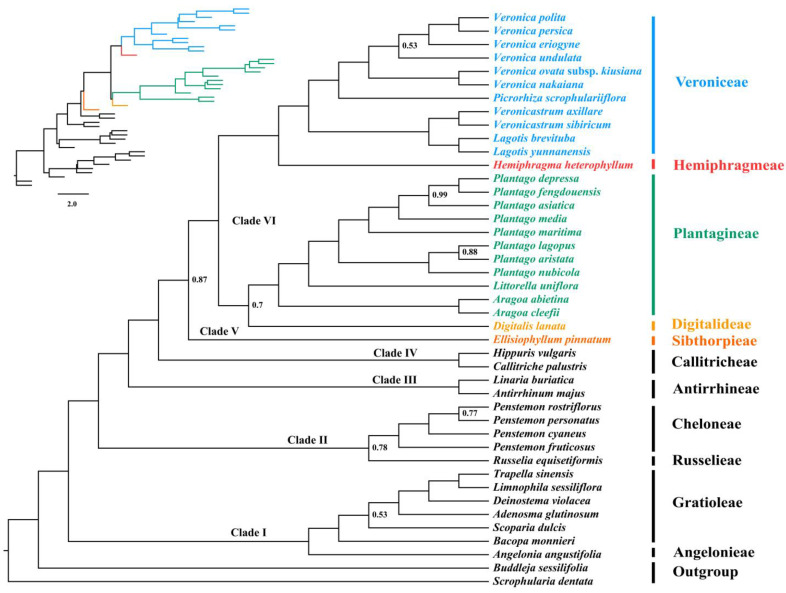
Species tree of Plantaginaceae reconstructed by ASTRAL based on 68 plastid protein-coding genes. Values at nodes represent local posterior probabilities (LPP) and all unlabeled nodes indicate high support values (LPP = 1.00). All 68 plastid gene trees of Plantaginaceae were shown in Appendix A.

**Figure 6 biology-12-00263-f006:**
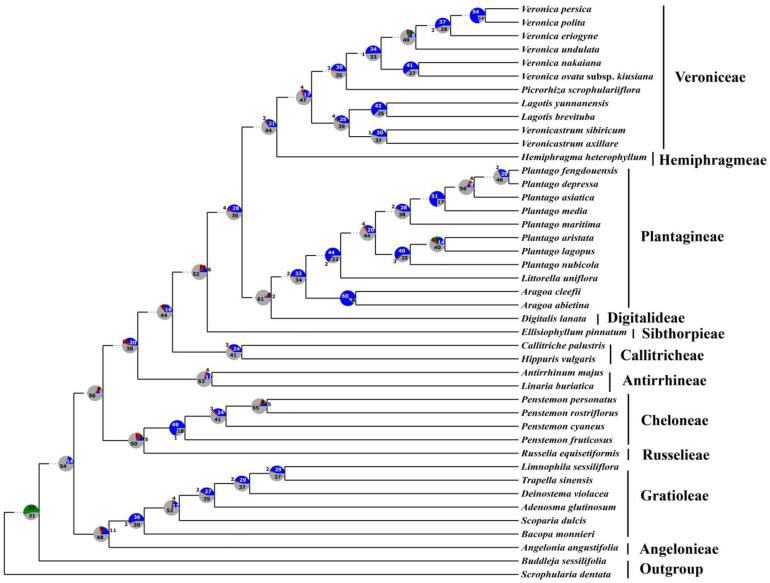
Summary of gene-tree concordance and conflict based on the PhyParts analysis of the 68 plastid gene trees mapped against the species tree (ASTRAL). The pie charts show the proportion and the numbers of genes in concordance (blue), conflict (green = a single dominant alternative; red = all other conflicting trees), and without information (grey) at each node in the species tree.

**Table 1 biology-12-00263-t001:** Comparison of plastome features among Plantaginaceae plants.

Tribes	Species	GC Content (%)	Size (bp)				Gene Number (Unique)			
			Genome	LSC	IR	SSC	Total	PCGs	rRNA	tRNA
Angelonieae	*Angelonia angustifolia*	37.7	154,316	84,110	27,128	15,950	112	78	4	30
Antirrhineae	*Antirrhinum majus*	37.9	152,606	83,205	25,734	17,933	114	80	4	30
	*Linaria buriatica*	37.8	150,665	81,766	25,648	17,603	114	80	4	30
Callitricheae	*Callitriche palustris*	37.8	150,138	81,432	25,667	17,372	114	80	4	30
	*Hippuris vulgaris*	37.6	152,763	82,983	25,743	18,294	114	80	4	30
Cheloneae	*Penstemon cyaneus*	37.9	152,604	83,724	25,534	17,812	114	80	4	30
	*Penstemon fruticosus*	37.9	152,704	83,684	25,599	17,822	114	80	4	30
	*Penstemon personatus*	37.9	152,602	83,728	25,528	17,818	114	80	4	30
	*Penstemon rostriflorus*	37.9	152,598	83,651	25,564	17,819	114	80	4	30
Digitalideae	*Digitalis lanata*	38.6	153,108	83,936	25,743	17,686	114	80	4	30
Gratioleae	*Deinostema violacea*	37.5	154,280	85,713	25,179	18,209	114	80	4	30
	*Adenosma glutinosum*	37.5	154,220	84,820	25,636	18,128	114	80	4	30
	*Bacopa monnieri*	37.6	152,495	83,765	25,668	17,394	114	80	4	30
	*Limnophila sessiliflora*	37.4	152,395	83,163	25,545	18,142	114	80	4	30
	*Scoparia dulcis*	37.5	153,701	85,029	25,273	18,126	113	79	4	30
	*Trapella sinensis*	37.4	152,297	83,830	25,010	18,447	114	80	4	30
Hemiphragmeae	*Hemiphragma heterophyllum*	38.1	152,707	83,268	25,808	17,823	114	80	4	30
Plantagineae	*Littorella uniflora*	39.0	130,833	77,490	21,404	10,535	103	69	4	30
	*Aragoa cleefii*	38.3	150,285	81,865	25,347	17,726	113	79	4	30
	*Aragoa abietina*	38.2	150,320	81,899	25,347	17,727	113	79	4	30
	*Plantago lagopus*	38.3	150,088	82,574	24,542	18,430	113	79	4	30
	*Plantago nubicola*	38.2	151,611	83,657	24,955	18,044	113	79	4	30
	*Plantago maritima*	38.6	158,358	82,223	33,735	8665	114	79	4	31
	*Plantago aristata*	38.4	149,910	82450	24,582	18,296	113	79	4	30
	*Plantago media*	38.0	164,130	82,757	38,398	4577	113	79	4	30
	*Plantago depressa*	38.0	164,617	82,933	38,388	4908	113	79	4	30
	*Plantago fengdouensis*	38.0	164,976	82,972	38,644	4716	113	79	4	30
	*Plantago asiatica*	38.1	165,045	82,964	38,724	4633	113	79	4	30
Russelieae	*Russelia equisetiformis*	38.2	153,337	83,844	25,738	18,017	114	80	4	30
Sibthorpieae	*Ellisiophyllum pinnatum*	38.2	152,424	83,232	25,787	17,618	114	80	4	30
Veroniceae	*Veronica polita*	37.9	150,191	81,847	25,465	17,414	113	79	4	30
	*Veronica persica*	37.9	150,198	81,850	25,465	17,418	113	79	4	30
	*Veronica eriogyne*	38.0	151,083	82,302	25,666	17,449	113	79	4	30
	*Veronica undulata*	38.1	151,178	82,644	25,566	17,402	114	80	4	30
	*Veronica ovata*	38.0	152,249	83,187	25,679	17,704	114	80	4	30
	*Veronica nakaiana*	37.9	152,319	83,195	25,711	17,702	114	80	4	30
	*Picrorhiza scrophulariiflora*	38.1	152,643	83,191	25,829	17,794	114	80	4	30
	*Veronicastrum axillare*	38.3	152,691	83,559	25,765	17,602	114	80	4	30
	*Veronicastrum sibiricum*	38.3	152,930	83,616	25,757	17,800	114	80	4	30
	*Lagotis brevituba*	38.3	152,967	83,740	25,691	17,845	114	80	4	30
	*Lagotis yunnanensis*	38.4	152,979	83,641	25,677	17,794	114	80	4	30

## Data Availability

The sequence data generated in this study are available in GenBank of the National Center for Biotechnology Information (NCBI) under the access numbers: OQ129603–OQ129608.

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
