# Peer review of "Plastid Phylogenomic Insights into the Inter-Tribal Relationships of Plantaginaceae"

_biology, 2023, doi:10.3390/biology12020263_

Round 1
Reviewer 1 Report
Plantaginaceae is a large family with great morphological diversity. The manuscript investigated inter-tribal relationships in Plantaginaceae based on plastomes.The results reconstructed the relationships among some tribals.But a few questions need to be clarified.
1、Why didn’t this research include Globularieae?
2、What is the detailed informations of every tribals, such as how many genera have in every tribals, since only one species was used in some tribals, and although four species were collected from Cheloneae, but all of them from genus Penstemon. If the intra-tribal and intra-genus variations should be considered.
3、Since 68 plastid gene trees were inferred separately, each trees should be supplemented in supplementary file. The consistent trees and conflicting trees should be presented with statistical data and the results should be discussed.
Reviewer 2 Report
In the manuscript "Plastid phylogenomics insights into the inter-tribal relationships of Plantaginaceae" by Xie et al. the authors present the plastid genome sequence of six Plantaginaceae species and compared them with 35 publicly available Plantaginaceae from Genebank. The authors infer phylogenies by concatenating the protein coding sequences of the plastomes and undertook a phylogenomic analysis using ASTRAL-III. The authors also perform an analysis to study congruence/conflict between the phylogenies of individual genes to those of the inferred species tree. This las analysis was performed with PhyParts.
The authors were able to reconstruct well supported phylogenies by the two approaches. Moreover, the topology of the tree inferred by concatenating protein coding genes into a single matrix and the topology inferred with ASTRAL were highly similar (only differed in a few terminal branches within Veronica and Penstemon).
It is interesting to see the large proportion of genes that support, conflict or were uninformative for several nodes in the species tree phylogeny (Figure 6). In particular, it is interesting to see that there are three nodes were there are more genes supporting conflicting topologies ((Plantaginae, Digitalidae),Sibthorpiae); (Chelonae, Russeliae).
Besides the genome sequence of the six plastids and the proposed species phylogeny of Plantaginaceae, the most valuable result is the analysis of congruence/conflict between the phylogenies of individual genes and the inferred species tree. I think the authors should refer more to this analysis when discussing the evolution of Plantaginaceae and previous work on this subject (see the next two paragraphs below).
For instance, regarding the sister relationship of Russelieae and Chelonae in the discussion section, I think the authors overemphasize the result of the tree in Figure 4. As shown in Figures 5 and 6, only 5 genes support this topology versus 13 in conflict. Also, in the sentence "Unlike most previous studies, our analyses favor a sister-group relationship between Plantagineae and Digitalidae..." the authors should consider that only 2 genes support this branching while 5 are in conflict.
More comments...
In Figures 4 and 5 a small tree is shown to the left of the main tree. It would be nice if the authors color the branches of the small tree according to the colors of the large tree. By this, the readers would identify which clades correspond between the trees.
A small chart showing this relationship "As shown in Figure 6, due to the lack of sufficient phylogenetic signal [40], most plastid genes are considered uninformative for nodes (BS < 70%) in species-tree phylogeny" would be appreciated. In fact, a chart showing the relationship of all support measures and the proportion of genes in concordance, conflict and uninformative would be highly appreciated.
The following sentence is a bit difficult to understand: "Notably, some nodes with numbers of conflicting genes close to or more than that of concordant genes seemed to have relatively low support in the species tree."
In the methods section, did the authors aligned protein coding genes taking into consideration codon structure? According to what is written, they did not. Please specify.
Finally, it would be very interesting if the authors map morphological characters mentioned in the Discussion section into the species tree. This would greatly improve the understanding of the manuscript for the non specialist and would show clearly the implication of the inferred species tree for morphological evolution. I think that this is not mandatory, but it would improve the article.
